# The burden of non-disabled frailty and its associated factors among older adults in Bangladesh

Sabuj Kanti Mistry[1,2,3]*, A. R. M. Mehrab Ali[1], Uday Narayan Yadav[2,4], Saruna Ghimire[5], Afsana Anwar[6], Md. Nazmul Huda[1,7], Fouzia Khanam[8], Rashidul Alam Mahumud[9], Ateeb Ahmad Parray[10], Shovon Bhattacharjee[11], David Lim[12], Mark Fort Harris[13]

1 ARCED Foundation, Dhaka, Bangladesh, 2 School of Population Health, University of New South Wales, Sydney, Australia, 3 Department of Public Health, Daffodil International University, Dhaka, Bangladesh, 4 National Centre for Epidemiology and Population Health, The Australian National University, Canberra, ACT, Australia, 5 Department of Sociology and Gerontology and Scripps Gerontology Center, Miami University, Oxford, OH, United States of America, 6 Rohingya Response Crisis, World Vision Bangladesh, Cox's Bazar, Bangladesh, 7 Discipline of Psychiatry and Mental Health, School of Clinical Medicine, University of New South Wales, Sydney, Australia, 8 Department of Public Health, North South University, Dhaka, Bangladesh, 9 NHMRC Clinical Trials Centre, Faculty of Medicine and Health, The University of Sydney, Camperdown, NSW, Australia, 10 Department of International Health, Johns Hopkins Bloomberg School of Public Health, Johns Hopkins University, Baltimore, MD, United States of America, 11 Biosecurity Program, The Kirby Institute, University of New South Wales, Sydney, Australia, 12 University of Technology Sydney, Ultimo, NSW, Australia, 13 Centre for Primary Health Care and Equity, University of New South Wales, Sydney, Australia

* smitra411@gmail.com

**Data Availability Statement:** All relevant data are within the paper and its Supporting Information files.

## Abstract

### Objective

The present study aims to measure the prevalence of non-disabled frailty and its associated factors among Bangladeshi older adults.

### Methods

This cross-sectional study was conducted during September and October 2021 among 1,045 Bangladeshi older adults (≥60 years). Telephone interviews, using a semi-structured questionnaire, were undertaken to collect data on participants' characteristics and level of frailty. The non-disabled frailty was measured using the 'Frail Non-Disabled (FiND)' questionnaire. A multinomial logistic regression model assessed the factors associated with frailty among the participants.

### Results

Around a quarter of the participants (24.8%) were frail. The multinomial regression analysis showed that older participants aged ≥80 years (RRR = 3.23, 95% CI: 1.41–7.37) were more likely to be frail compared to participants aged 60–69 years. Likewise, the participants living in a large family with ≥4 members (RRR = 1.39, 95% CI: 1.01–1.92) were more likely to be frail compared to those living in smaller families. Also, participants having memory or

**Funding:** The author(s) received no specific funding for this work.

**Competing interests:** The authors have declared that no competing interests exist.

concentration problems (RRR = 1.56, 95% CI: 1.12–2.17) were more likely to be frail compared to those who were not suffering from these problems. Moreover, participants whose family members were non-responsive to their day-to-day assistance (RRR = 1.47, 95% CI: 1.06–2.03) were more likely to be frail compared to those whose family members were responsive. Furthermore, participants who were feeling lonely (RRR = 1.45, 95% CI: 1.07–1.98) were more likely to be frail than their counterparts who were not feeling lonely.

## Conclusions

The findings of the present study suggest developing tailored interventions to address the burden of frailty among the older populations in Bangladesh. In particular, providing long-term care and health promotion activities can be of value in preventing frailty and reducing adverse health outcomes among this vulnerable population group.

## Introduction

The world is rapidly transitioning into an ageing society. According to the World Health Organization (WHO), one in six individuals worldwide, accounting for 1.4 billion people will be 60 years or older by 2030, and that population number will be 2.1 billion by 2050 [1]. Low- and middle- income countries (LMICs) are projected to have two-thirds of their population aged 60 or older by 2050 [1]. In Bangladesh, reflecting this global demographic shift, the total number of older adults aged 60 years and above is expected to increase from 8 million to 44 million by 2050 [2]. This demographic change has resulted in an increased burden of conditions common in older age, including various non-communicable diseases (NCDs) and frailty [3]; the latter is now recognized as an emerging concern among older adults in South Asian countries, including Bangladesh [4].

According to Fried and colleagues [5], frailty is a condition where three out of five phenotypic criteria of an individual are met: shrinkage (weight loss), exhaustion, weaknesses, low gait speed, and low physical activity. Frailty predicts many long-term health outcomes, disability and mortality [6] and interferes with one's ability to live independently [7]. Frailty is common in older age due to biological consequences that result in the deterioration of physical, cognitive, and psychological functions [8]. This condition manifests as a state of increased vulnerability and decreased resilience, resulting in decreased physical activity, diminished cognitive performance, and compromised nutritional status [7].

There is evidence indicating a high burden of frailty in LMICs. A systematic review and meta-analysis of 56 studies conducted in LMICs documented that the prevalence of frailty and prefrailty among older adults aged 60 years and above was 17.4% and 49.3% respectively [9]. The prevalence of frailty was high among older adults from South Asian countries, ranging between 25% and 65%. For instance, a Nepalese study reported that 65% of participants had frailty without disability [6]. In India, 26% of older adults were frail, while 63.6% were prefrail, and only 10.4% were non-frail [10]. Studies conducted in Pakistan [11] and Sri Lanka [12] found that 55.4% and 15.2% of older adults were frail, with 44.6% and 48.5% being prefrail, respectively. These studies also identified several determinants of frailty among older adults, such as age, education, occupation, out-of-pocket health care cost, depression, and lack of physical activities and care from family members [6, 10–12].

Limited evidence exists on the prevalence and determinants of frailty among older adults in Bangladesh. The only study on frailty among Bangladeshi older adults showed that 61.6% of

the participants had moderate to severe levels of frailty [13]. However, this study was conducted in a small geographical area of the country and used the 30-indicator Frailty Index scale that does not measure non-disabled frailty. Considering the fact that there is high burden of NCDs and limited access to health service facilities in Bangladesh, we anticipate a high prevalence of frailty among older adults. Therefore, the present study was undertaken to estimate the prevalence of non-disabled frailty and its determinants among Bangladeshi older adults.

## Materials and methods

### Study design and participants

This cross-sectional study was conducted during September and October 2021. Previously, between 2016–2020, the ARCED (Aureolin Research, Consultancy, and Expertise Development) Foundation conducted ten different community-based studies in Bangladesh and established a registry using demographic information of participants from these studies, which served as the sampling frame for the current study. Notably, this sampling frame included households from all eight administrative divisions of Bangladesh. Considering a 50% prevalence with a 5% margin of error, 95% confidence level, and 90% power, a sample size of 1096 was calculated. However, only 1045 of 1096 approached participants responded to the study resulting in an overall response rate of approximately 95%. Based on the population distribution of older people by geography in Bangladesh, we adopted a probability proportionate to size (of the eight administrative divisions) approach to determine the size of older people in each division [14]. The inclusion criterion was the minimum age of 60 years of the participants. Exclusion criteria were suffering from any adverse mental conditions i.e., clinically diagnosed schizophrenia, bipolar mood disorder, dementia/cognitive impairment, and inability to communicate.

### Measures

**Outcome measure.**   The outcome of interest was non-disabled frailty, measured using the 'Frail Non-Disabled (FiND) questionnaire'[15]. The FiND questionnaire is designed to measure non-disabled frailty among the older population and was previously used among older adults in Nepal in a similar setting [6]. The FiND questionnaire contains five questions under two sections. The first two questions on mobility belong to the disability section, and the latter three on weight loss, exhaustion, and physical activity belong to the frailty section. Each five items are coded as 0 and 1, indicating negative and affirmative responses, respectively. Participants are categorized into one of three categories, i.e., "disabled," "frail," and "robust." If a participant responds affirmatively to at least one of the two disability items, they are classified as disabled. Frailty is when a participant responds negatively to both disability questions but affirmatively to at least one frailty item. A participant is classified as "robust" when the sum of five items is zero, i.e., indicating the absence of both frailty or disability." A Cronbach α of 0.75 in the current study indicated the scale is reliable among our study population.

**Explanatory variables.**   A literature review guided the selection of explanatory variables for our stusy [6, 10–12, 16]. Explanatory variables considered in this study were administrative divisions (Barishal, Chattogram, Dhaka, Mymensingh, Khulna, Rajshahi, Rangpur, Sylhet), age (categorized as 60–69, 70–79, and ≥80), gender (male/female), marital status (married and without a partner; latter included widowed, separated and never married), formal schooling (with and without formal schooling), family size (≤4 or >4), family monthly income in Bangladeshi Taka (BDT) (<5,000, 5,000–10,000, >10,000), residence (urban/rural), current occupation (employed, unemployed or retired), living arrangement (living alone or with family), walking distance to the nearest health center (<30 min/≥30 min), memory or concentration

problems (no problem/low memory or concentration), suffering from at least one chronic conditions (yes/no), a perception that family members are non-responsive to their day-to-day assistance (yes/no), and feeling of loneliness (yes/no).

Self-reported information on the participant's ever diagnosis of chronic conditions (i.e., arthritis, hypertension, heart diseases, stroke, hypercholesterolemia, diabetes, chronic respiratory diseases, chronic kidney disease, and cancer) was collected as dichotomized yes/no responses. To explore if participants had any memory or concentration problems, we asked the following question: 'Do you have any memory (remembering things properly) or concentration (focusing/concentrating properly while doing any action) problem?' Loneliness was measured using the 3-item UCLA Loneliness scale [17]. Each item on the scale was measured as a yes/no question. The three items included: how often do you feel: (i) lack companionship, (ii) left out, and (iii) isolated in the last two weeks? Each item in the scale was measured in terms of 3-item Likert responses: hardly ever (1 point), some of the time (2 points), and often (3 points). Participants were classified as lonely if they answered 'some of the time' or 'often' to any of the three items [17].

## Data collection tools and techniques

Telephone interviews were conducted using a pre-tested semi-structured questionnaire, and data were collected electronically using SurveyCTO mobile app (https://www.surveycto.com/). Ten research assistants were recruited as surveyors based on their previous experiences administering health surveys on the electronic platform. The research assistants were trained extensively via Zoom meetings for three full days by the investigators (SKM, AMA, UNY) before beginning the data collection.

The English version of the questionnaire was first translated into Bengali and then back-translated to English by two researchers (SKM and AMA) to ensure the content's construct and face validity. The questionnaire was then piloted among a small sample (n = 10) of older adults for cultural validation of contents and refinement of the questions. The piloted questionnaire did not receive any corrections/suggestions and thus was used as it is for the data collection. Each interview took around half an hour.

## Statistical analyses

Descriptive analyses (i.e., frequency and percentage) were performed to explore the distribution of the sociodemographic characteristics of the respondents. To explore the factors independently associated with non-disabled frailty, multinomial logistic regression models were run, adjusting for important confounders. Covariates for adjustment were identified using the backward elimination method based on the Akaike information criterion (AIC). The initial model was run with all potential covariates listed in Table 1, and based on the AIC, age, formal schooling, family size, residence, problems with memory or concentration, loneliness, and non-responsive family members were retained. The final model was executed including these variables. The final model was also tested for sensitivity using the bootstrapping approach by resampling observations with 10,000 replications. The Relative Risk Ratio (RRR) and associated 95% confidence interval (95% CI) for the "disabled" and "frail" categories are provided with reference to the "robust" category in Table 2. All analyses were performed using the statistical software package Stata (Version 14.0).

## Ethics approval

The study was approved by the Institutional Review Board of the Institute of Health Economics, the University of Dhaka, Bangladesh (Ref: IHE/2020/1037). Verbal informed consent was

**Table 1. Characteristics of study participants (N = 1045).**

| Characteristics | | N | % |
|---|---|---|---|
| Administrative division | | | |
| | Barishal | 146 | 14.0 |
| | Chattogram | 98 | 9.4 |
| | Dhaka | 172 | 16.5 |
| | Mymensingh | 69 | 6.6 |
| | Khulna | 198 | 19.0 |
| | Rajshahi | 145 | 13.9 |
| | Rangpur | 161 | 15.4 |
| | Sylhet | 56 | 5.4 |
| Residence | | | |
| | Urban | 182 | 17.4 |
| | Rural | 863 | 82.6 |
| Age (in years) | | | |
| | 60–69 | 790 | 75.6 |
| | 70–79 | 201 | 19.2 |
| | $\geq$ 80 | 54 | 5.2 |
| Gender | | | |
| | Male | 620 | 59.3 |
| | Female | 425 | 40.7 |
| Marital status | | | |
| | Married | 799 | 76.5 |
| | [1]Without partner | 246 | 23.5 |
| Formal schooling | | | |
| | No formal schooling | 540 | 51.7 |
| | Formally schooled | 505 | 48.3 |
| Family size | | | |
| | $\leq$4 | 347 | 33.2 |
| | >4 | 698 | 66.8 |
| [2]Family monthly income (BDT) | | | |
| | <5000 | 121 | 11.6 |
| | 5000–10000 | 469 | 44.9 |
| | >10000 | 455 | 43.5 |
| Current occupation | | | |
| | Employed | 407 | 39.0 |
| | Unemployed/retired | 638 | 61.1 |
| Living arrangement | | | |
| | Living with family | 992 | 94.9 |
| | Living alone | 53 | 5.1 |
| Walking distance to the nearest health centre | | | |
| | <30 minute | 581 | 55.6 |
| | $\geq$30 minutes | 464 | 44.4 |
| Problems with memory or concentration | | | |
| | No problem | 676 | 64.7 |
| | Low memory or concentration | 369 | 35.3 |
| Suffering from at least one chronic condition | | | |
| | No | 447 | 42.8 |
| | Yes | 598 | 57.2 |

(*Continued*)

**Table 1.** (Continued)

| Characteristics | | N | % |
|---|---|---|---|
| Family members non-responsive to their day-to-day assistance | | | |
| | No | 738 | 70.6 |
| | Yes | 307 | 29.4 |
| Feeling of loneliness | | | |
| | No | 568 | 54.4 |
| | Yes | 477 | 45.7 |

[1]Without partner group includes widowed, separated and never married

[2]One USD = 85.75 Bangladeshi Taka (BDT).

sought from the participants before administering the survey. Participation in the survey was voluntary and participants did not receive any compensation for their time.

## Patient and public involvement

Patients and/or the public were not involved in developing research questions, designing and conducting the study, and disseminating the results.

**Table 2. Factors associated with non-disabled frailty and disability among the participants (N = 1045).**

| Characteristics | Disability | | | Frailty | | |
|---|---|---|---|---|---|---|
| | RRR[1] | 95% CI | *P* | RRR | 95% CI | *P* |
| Age (years) | | | | | | |
| 60–69 | *Ref* | | | *Ref* | | |
| 70–79 | 2.56 | 1.72–3.80 | <0.001 | 1.08 | 0.72–1.62 | 0.718 |
| ≥ 80 | 10.00 | 4.64–21.56 | <0.001 | 3.23 | 1.41–7.37 | 0.005 |
| Formal schooling | | | | | | |
| No formal schooling | *Ref* | | | *Ref* | | |
| Having formal schooling | 0.75 | 0.53–1.06 | 0.109 | 0.84 | 0.62–1.14 | 0.270 |
| Family size | | | | | | |
| ≤4 | *Ref* | | | *Ref* | | |
| >4 | 1.25 | 0.87–1.80 | 0.235 | 1.39 | 1.01–1.92 | 0.047 |
| Residence | | | | | | |
| Urban | *Ref* | | | *Ref* | | |
| Rural | 0.61 | 0.39–0.96 | 0.032 | 0.69 | 0.46–1.01 | 0.059 |
| Problems with memory or concentration | | | | | | |
| No problem | *Ref* | | | *Ref* | | |
| Low memory or concentration | 2.73 | 1.91–3.91 | <0.001 | 1.56 | 1.12–2.17 | 0.002 |
| Feeling of loneliness | | | | | | |
| No | *Ref* | | | *Ref* | | |
| Yes | 1.47 | 1.03–2.08 | 0.032 | 1.45 | 1.07–1.98 | 0.017 |
| Family members non-responsive to their day-to-day assistance | | | | | | |
| No | *Ref* | | | *Ref* | | |
| Yes | 1.09 | 0.75–1.58 | 0.663 | 1.47 | 1.06–2.03 | 0.020 |

[1]RRR = Relative Risk Ratio; Note: The adjusted model contained all variables in Table 2.

## Results

### Characteristics of the participants

The sociodemographic characteristics of the participants are presented in Table 1. The majority of the participants were aged 60–69 years (75.6%), males (59.3%), and married (76.5%). More than half of the participants (51.7%) had no formal schooling, 66.8% belonged to a large family of more than four members, 82.6% lived in rural areas, and 61.1% were unemployed or retired during the survey period. Furthermore, 35.3% had problems with memory or concentration, 57.2% suffered from at least one non-communicable chronic condition, 29.4% had their family members non-responsive to their day-to-day assistance, and 45.7% felt lonely (Table 1).

### Prevalence of non-disabled frailty

Fig 1 shows the prevalence of non-disabled frailty, disability, and robustness by age and gender using the FiND questionnaire. We found that around one-fourth (24.8%) of the total participants (n = 1045) had non-disabled frailty, while 19.7% had a disability. No statistically significant differences were observed in the prevalence of frailty and disability among participants in terms of age and sex.

### Factors associated with non-disabled frailty

Table 2 shows the factors associated with non-disabled frailty and disability (with reference to the robust category) in the adjusted multinomial logistic regression model. Although findings for both non-disabled frailty and disability are presented in Table 2, given our research question, we interpret here the findings relevant to non-disabled frailty only. In the adjusted model, compared to the youngest age group (60–69 years), the oldest group (≥80 years) had more than three times higher risk of frailty (RRR = 3.23, 95% CI: 1.41–7.37). Likewise, participants living in large families with four or more members (RRR = 1.39, 95% CI: 1.01–1.92) had

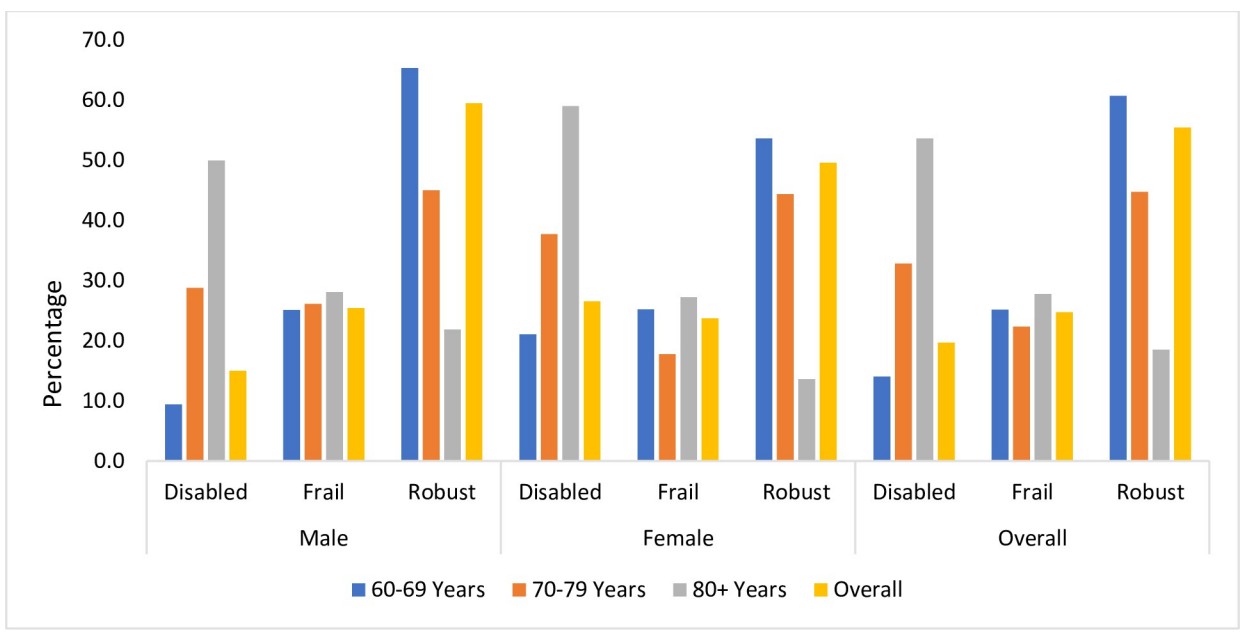

**Fig 1. Prevalence of non-disabled frailty by age and gender (n = 1045).**

39% higher risk of being frail compared to those living in small families. Also, participants whose family members were non-responsive to their day-to-day assistance (RRR = 1.47, 95% CI: 1.06–2.03) had 47% higher risk of being frail than those who had family members responsive to their day-to-day assistance. Participants with low memory and concentration problems had a 56% higher risk of being frail (RRR = 1.56, 95% CI: 1.12–2.17) than those without such problems. Moreover, participants reporting loneliness were significantly more likely to be frail than their counterparts (RRR = 1.45, 95% CI: 1.07–1.98). Sensitivity analysis suggested the results to be robust in quantifying consistent associations (S1 Table).

## Discussion

The present study revealed that 24.8% of the participants had non-disabled frailty and factors such as age, family size, memory or concentration problems, non-responsive family members and loneliness were associated with frailty. Our study complement to the limited literature on frailty among South Asian older adults, specifically from Bangladesh.

Our reported 24.8% prevalence of frailty among older adults is similar to studies in India (26%) [10] and Italy (23.2%) [18]. Likewise, a systematic review of studies conducted in developed countries on frailty among community-dwelling older adults reported a pooled prevalence of 10.7% [19]. In contrast, studies from two other South Asian countries, Nepal and Pakistan, have reported a higher prevalence of frailty compared to that of our study [6, 11]. In a study from rural Nepal, 65% of participants were frail [6], while in the Pakistani study, 55.4% were frail [11]. The prevalence of frailty varies across studies, which may be attributable to several factors, including different measurement tools, heterogeneity in participants' ages, people's living standard, and the burden of NCDs.

In our study, older people aged $\geq$ 80 were more likely to be frail than those aged 60–69 years, which aligns with existing literature [20, 21]. Several biological and physiological changes accompanying the ageing process and a high burden of NCDs among older adults explain their increased likelihood of frailty [22]. Like many other countries of the world (e.g., India, China and Indonesia), Bangladesh is also facing an epidemiological transition, resulting in a higher burden of NCDs [23], which might have contributed to a higher level of frailty among its older population. Additionally, frailty was found to be associated with cognitive impairment, such as having problems with memory and concentration, in the present study and in some prior studies [20, 21], which may also add to the burden among older adults.

Consistent with the existing literature [13], we found that older adults who were living in large families and reporting their family members as non-responsive to their day-to-day assistance had a higher likelihood of frailty. Several contextual factors together explain the observed findings between family size and frailty. Social safety net interventions such as insurance and pensions are largely unavailable for older people in Bangladesh. Thus, at an older age, many Bangladeshi people financially, emotionally, and physically depend on their adult family members and receive care from them. Evidence suggests that Bangladeshi large families often suffer a lot in satisfying the daily needs of their family members and older family members also receive suboptimal care [24]. Consequently, older adults may feel that the family members are non-responsive to their needs. The financial constraints could also be a barrier to good nutrition and access to health care for older parents [25]. All these issues can compromise older people's quality of life and make them vulnerable to frailty and other adverse health outcomes [26, 27].

Like other studies [17, 28], this study also found an association between frailty and loneliness. Evidence suggests that high levels of loneliness result in declined mobility and increased difficulties with daily living activities [17]. Lonely people are more likely to be inactive, which

increases the risk of physical frailty [17, 28]. We acknowledge that in Bangladesh, where multi-generational households are common, living alone is not typical. Culturally, in Bangladesh, older people are attached to families, living with children and grandchildren [29]. Although living in a large multigenerational family offers social support, loneliness may be partly due to losing a spouse and friends, which is common in old age. Second, migration is very high in Bangladesh [30]. Even in a multigenerational family setting, a void is left by migrant children. Evidence suggests that the left-behind parents suffer the emotional cost of migration [31]. For some, loneliness may also be created due to neglect from family members which was more evident in our study as about 30% reported their family members were not responsive to their needs. As discussed in the previous paragraph, the financial constraint and demand to meet competing needs of multiple family members may result in sub-optimal care for the older family members and thus a feeling of neglect [24]. Recently, change in family dynamics is also noted in Bangladesh where nuclear family is being more popular and in some cases, older adults live with paid maids, separated from their children [29]. Older people's limited social support from their children/family members and being separated from their children increase the likelihood of depression, loneliness, and cognitive decline [32], all of which can contribute to developing frailty.

## Implications for policy and practice

Due to the demographic transition resulting in a burgeoning number of older adults in Bangladesh [2], the prevalence of frailty is projected to increase. Health and social care services are meager for the older population in Bangladesh, and the existing family care is challenged. The Government of Bangladesh has introduced National Policy on Ageing (2007), the Parents Care Act (2013), and Old Age Allowance (OAA) program [33], but on a limited scale. There is a lack of a national policy or strategy focusing on old age health and social care issues in Bangladesh, which is critical to addressing frailty and disability. Due to the absence of healthcare insurance support and resulting high out-of-pocket costs, preventive services are underutilized [3]. Therefore, introducing health insurance could be crucial to access preventive services and ensure early diagnosis, treatment and delayed progression of frailty. Providing comprehensive supportive care, including multidisciplinary care services, individualized care, and long-term follow-up to address the holistic needs of older adults, would be particularly important to address frailty. Strengthening primary care services, including promoting physical activities and ensuring proper nutrition and health education, are also proven effective in managing frailty.

## Strengths and limitations of the study

This study's generalizability is one of its strengths, as it collected data from all eight divisions of Bangladesh. Other notable methodological strengths include the large sample size, using a standard tool to measure outcomes, and data collection by trained and experienced surveyors. However, the cross-sectional design limits the ability to establish a causal association. Due to the paucity of literature on frailty in Bangladesh, the current study intended to quantify the burden and serve as a baseline for future studies. We recommend a longitudinal design in the future to better establish the predictors for frailty in this population. Although we used a validated scale for measuring frailty, it was not explicitly validated among Bangladeshi older adults. Thus, future studies also have an opportunity to validate the FiND questionnaire in Bangladesh. Given the self-reported nature of the survey, we also anticipate measurement bias in our study. Finally, our study is limited to quantitative analysis, which helped us to understand the potential association. However, to better understand the underlying reasons for

these associations, specifically the observed relationships between family and frailty, we recommend a future qualitative study.

## Conclusion

The present study found one in four older adults had non-disabled frailty in Bangladesh. Specifically, those relatively older, living in large families, suffering from cognitive problems or loneliness, and not properly cared for by their family members were more vulnerable to frailty. The findings suggest the need for developing tailored interventions including early screening of frailty at primary health care services to address emerging burden of frailty. Moreover, providing long-term care and relevant health promotion services such as engagement in physical activities, providing nutritional support, and delivering older people friendly services may effectively prevent frailty and reduce associated long-term adverse health outcomes.

## Supporting information

**S1 Table. Factors associated with non-disabled frailty and disability among the participants (N = 1045).**
(DOCX)

**S1 File. Dataset used for the analysis in the current study.**
(DTA)

## Acknowledgments

We acknowledge the role of Sadia Sumaia Chowdhury, Programme Manager, ARCED Foundation and Md. Zahirul Islam, Project Associate, ARCED Foundation, for their support in data collection for the study.

## Author Contributions

**Conceptualization:** Sabuj Kanti Mistry, A. R. M. Mehrab Ali, Uday Narayan Yadav.

**Data curation:** Sabuj Kanti Mistry.

**Formal analysis:** Sabuj Kanti Mistry.

**Investigation:** Sabuj Kanti Mistry, A. R. M. Mehrab Ali.

**Methodology:** Sabuj Kanti Mistry, A. R. M. Mehrab Ali, Uday Narayan Yadav.

**Software:** Sabuj Kanti Mistry.

**Supervision:** Sabuj Kanti Mistry, A. R. M. Mehrab Ali.

**Writing – original draft:** Sabuj Kanti Mistry, A. R. M. Mehrab Ali, Uday Narayan Yadav, Saruna Ghimire, Afsana Anwar, Md. Nazmul Huda, Fouzia Khanam, Rashidul Alam Mahumud, Ateeb Ahmad Parray, Shovon Bhattacharjee, David Lim.

**Writing – review & editing:** Mark Fort Harris.

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
