## [Decision Letter · Decision Letter 0]

30 May 2023

PONE-D-23-00513

The burden of non-disabled frailty and its associated factors among older adults in Bangladesh

PLOS ONE

Dear Dr. Mistry,

Thank you for submitting your manuscript to PLOS ONE. After careful consideration, we feel that it has merit but does not fully meet PLOS ONE’s publication criteria as it currently stands. Therefore, we invite you to submit a revised version of the manuscript that addresses the points raised during the review process.

Detailed comments from both reviewers and especially reviewer 2, should help the authors to improve the manuscript. Although detailed and many, the comments should not very difficult to address. I encourage the authors to address all the comments sincerely. 

We look forward to receiving your revised manuscript.

Kind regards,

Shahriar Ahmed, MBBS, MHE

Academic Editor

PLOS ONE

Reviewers' comments:

Reviewer's Responses to Questions

**Comments to the Author**

1. Is the manuscript technically sound, and do the data support the conclusions?

Reviewer #1: Partly

Reviewer #2: Partly

2. Has the statistical analysis been performed appropriately and rigorously? 

Reviewer #1: Yes

Reviewer #2: No

3. Have the authors made all data underlying the findings in their manuscript fully available?

Reviewer #1: Yes

Reviewer #2: Yes

4. Is the manuscript presented in an intelligible fashion and written in standard English?

Reviewer #1: Yes

Reviewer #2: No

5. Review Comments to the Author

Reviewer #1: The authors estimated both non-disabled frailty and disability using the FiND questionnaire but I am not clear on the reasons for the inclusion of the questions about disability or the inclusion in the results of disability and robustness.

The authors give some prominence to the male/female difference in non-disabled frailty yet the difference is small and does not reach statistical significance. I would recommend not giving this finding such prominence as it would be difficult to recommend policy based on this.

Minor issues: Conclusions in the abstract; should read "present" rather than "preset".

Figure 1; "Disable" should read "Disabled".

Figure 1, the Y=axis needs label (presumably %?)

Reviewer #2: External Reviewer

This research on non-disabled frailty among older adults is very important for health and well-being of elderly populations and for that I think this is an important study.

I have a few comments/questions that I think should be addressed properly.

Comments:

Abstract:

1. Results: Revision of the results section required for further clarity. Please check for grammatical issues also.

2. Conclusion: Your conclusion should be based on the findings of the study (Please revise). What do you mean by “comprehensive support care package”? Typographical errors (tense, spelling, etc).

Introduction:

3. Frailty suddenly appeared in the last sentence of first paragraph and the link between first and second paragraph not properly done. Revision required.

4. Last sentence of the second paragraph was hard to grasp on what you were trying to convey.

5. Last paragraph requires attention and significant revision. Sentences beginning with “Thus” appeared in almost everywhere.

6. There are typographical errors (tense, spelling, grammar) that could be fixed to improve the readability of the manuscript (Introduction, Methodology, Results and Discussion). Introduction part requires significant rearrangement. Rationale should be done properly based on the objective.

7. Referencing should be done properly.

Methods and materials:

8. Sampling framework hard to make any senses. Please revise

9. Sample size based on the “response rate” and overall “response rate “creating confusion. Please revise accordingly.

10. “The inclusion criterion was the minimum age of 60 years” of whom? Please check the whole manuscript for sentence making so that conveying messages make senses.

11. First sentence of the outcome measure is hard to grasp. Please revise.

12. Multiple grammatical errors in this section.

13. What is “robust”?

14. Explanatory variables- what do you mean by “without partner” in the “marital status (married/without a partner)” variable?

15. How many research assistants you had recruited for the study? Did you provide any training before the actual data collection? If yes, how long?

16. Data collection tools and techniques require significant revision.

17. Statistical analyses lack proper guidance what you have done in the paper.

18. Ethics approval: “All procedures performed in studies….” Did not understand? Second sentence failed to convey messages properly. Please revise.

Results:

19. Characteristics of the participants- Please revise with most important information that aligns with your objectives.

20. 19.7% were considered disabled (more than 200 people out of 1045!). This is huge number. Can you please share what types of disability you found in this study?

21. “perceiving family members to be non-responsive to their needs (RRR= 1.47, 95% CI: 1.06-2.03)….” what do you mean by “perceiving family members”?

22. What is “concentration”? Provide the definitions of “low memory” and “concentration”.

23. Please revise this statement- “participants who reported a feeling of loneliness had a significantly higher risk of frailty compared to those who did not feel so”.

Discussion:

24. Did not understand the first paragraph of the discussion section. Please revise to convey the messages properly.

25. “The discrepancies in frailty prevalence between the study and the literature may be attributable to methodological factors such as…” and after that you provided examples of determinants not the methodological factors. This paragraph requires significant revision.

26. “In our study, the odds of frailty were higher in the higher age group ….” In the results, you mentioned about RRR. Here you are talking about odds? Why, revise accordingly.

27. “Like many other countries of the world, Bangladesh is also facing an epidemiological transition” What are the countries you are talking about? Rich, poor, developing, developed?

28. “We found that older adults living in a large family and reported having their family members nonresponsive to their care had a higher chance of frailty. These two things are interlinked….” How they are interlinked? Did you analyze that? Did you mention that in the results?

29. Explanation of the fourth paragraph of the discussion section is not comprehensive and logical enough. Please provide scientific explanations with relevant references.

30. “After a certain period of life, depending financially, emotionally, and physically on children and receiving care from there is culturally expected and appropriate.” Don’t understand what you are trying to convey here?

31. Need more elaborations on limitations of the study.

Conclusion:

32. “Overall, the findings of the study suggest analyzing and understanding frailty and its associated components is an essential first step toward that preventive goal” What are the associated components you are trying to imply here?

33. Please revise the conclusion section for better understanding.

6. PLOS authors have the option to publish the peer review history of their article (what does this mean?). If published, this will include your full peer review and any attached files.

Reviewer #1: **Yes: **Calum Mattocks

Reviewer #2: No

---

## [Author Response · Author response to Decision Letter 0]

5 Jul 2023

Please see the attachment 'Review response_R1'

---

## [Decision Letter · Decision Letter 1]

17 Aug 2023

PONE-D-23-00513R1The burden of non-disabled frailty and its associated factors among older adults in BangladeshPLOS ONE

Dear Dr. Mistry,

Thank you for submitting your manuscript to PLOS ONE. After careful consideration, we feel that it has merit but does not fully meet PLOS ONE’s publication criteria as it currently stands. Therefore, we invite you to submit a revised version of the manuscript that addresses the points raised during the review process.

We believe that the first revision was comprehensive and was able to address most of the comments from the reviewers. However, the comments from reviewer 2 should not be very difficult to address and we believe that doing so will significantly improve the quality and acceptability of the manuscript. We look forward to receiving your prompt response.

We look forward to receiving your revised manuscript.

Kind regards,

Shahriar Ahmed, MBBS, MHE

Academic Editor

PLOS ONE

Journal Requirements:

Additional Editor Comments:

Dear authors, thank you for submitting the manuscript for consideration. We are happy to see that the overall quality of the manuscript has much improved during the first revision. We believe that you will be able to address the rest of the comments from one of the reviewers and we will be able to move forward with a final decision soon.

Reviewers' comments:

Reviewer's Responses to Questions

**Comments to the Author**

1. If the authors have adequately addressed your comments raised in a previous round of review and you feel that this manuscript is now acceptable for publication, you may indicate that here to bypass the “Comments to the Author” section, enter your conflict of interest statement in the “Confidential to Editor” section, and submit your "Accept" recommendation.

Reviewer #1: All comments have been addressed

Reviewer #2: All comments have been addressed

2. Is the manuscript technically sound, and do the data support the conclusions?

Reviewer #1: Yes

Reviewer #2: Partly

3. Has the statistical analysis been performed appropriately and rigorously? 

Reviewer #1: Yes

Reviewer #2: Yes

4. Have the authors made all data underlying the findings in their manuscript fully available?

Reviewer #1: Yes

Reviewer #2: Yes

5. Is the manuscript presented in an intelligible fashion and written in standard English?

Reviewer #1: Yes

Reviewer #2: Yes

6. Review Comments to the Author

Reviewer #1: (No Response)

Reviewer #2: Please see the comments provided in the pdf. The author need to address the comments properly in the revised version for further clarification.

7. PLOS authors have the option to publish the peer review history of their article (what does this mean?). If published, this will include your full peer review and any attached files.

Reviewer #1: **Yes: **Calum Mattocks

Reviewer #2: No

---

## [Author Response · Author response to Decision Letter 1]

22 Aug 2023

Review response is added as an attachement.

---

## [Decision Letter · Decision Letter 2]

13 Nov 2023

The burden of non-disabled frailty and its associated factors among older adults in Bangladesh

PONE-D-23-00513R2

Dear Dr. Mistry,

We’re pleased to inform you that your manuscript has been judged scientifically suitable for publication and will be formally accepted for publication once it meets all outstanding technical requirements.

Kind regards,

Shahriar Ahmed, MBBS, MHE, MPhil

Academic Editor

PLOS ONE

Additional Editor Comments (optional):

Dear authors,

Thank you for your patience with us. This manuscript can now be accepted with minor revisions, mostly related to language edits. We believe that you have benefitted from the review process although it took longer than we would like. However, we are happy that we have been able to recommend a decision to the Editor. We wish you best of luck.

Thank you!

Reviewers' comments:

Reviewer's Responses to Questions

**Comments to the Author**

1. If the authors have adequately addressed your comments raised in a previous round of review and you feel that this manuscript is now acceptable for publication, you may indicate that here to bypass the “Comments to the Author” section, enter your conflict of interest statement in the “Confidential to Editor” section, and submit your "Accept" recommendation.

Reviewer #1: All comments have been addressed

Reviewer #2: (No Response)

2. Is the manuscript technically sound, and do the data support the conclusions?

Reviewer #1: (No Response)

Reviewer #2: Yes

3. Has the statistical analysis been performed appropriately and rigorously? 

Reviewer #1: (No Response)

Reviewer #2: Yes

4. Have the authors made all data underlying the findings in their manuscript fully available?

Reviewer #1: (No Response)

Reviewer #2: Yes

5. Is the manuscript presented in an intelligible fashion and written in standard English?

Reviewer #1: (No Response)

Reviewer #2: Yes

6. Review Comments to the Author

Reviewer #1: (No Response)

Reviewer #2: (No Response)

7. PLOS authors have the option to publish the peer review history of their article (what does this mean?). If published, this will include your full peer review and any attached files.

Reviewer #1: No

Reviewer #2: No

---

## [Editor Report · Acceptance letter]

17 Nov 2023

PONE-D-23-00513R2 

The burden of non-disabled frailty and its associated factors among older adults in Bangladesh 

Dear Dr. Mistry:

I'm pleased to inform you that your manuscript has been deemed suitable for publication in PLOS ONE. Congratulations! Your manuscript is now with our production department. 

Kind regards, 

on behalf of

Dr. Shahriar Ahmed 

Academic Editor

PLOS ONE